# Construction and Validation of Simulation Models of Samples Made from 316L Steel by Applying Additive Technique

**DOI:** 10.3390/ma15186244

**Published:** 2022-09-08

**Authors:** Kamila Dąbrowska, Radosław Nowak, Przemysław Rumianek, Jarosław Seńko

**Affiliations:** Faculty of Automotive and Construction Machinery Engineering, Warsaw University of Technology, 02-524 Warsaw, Poland

**Keywords:** 3D printing, steel 316L, strength testing, finite element method, laminate 3D

## Abstract

The main aim of the study includes research concerning the strength of samples printed out of 316L steel in the form of laminates and the creation of reflective simulation models with regard to the results obtained during the research. In addition, the tests addressed the effect of the arrangement of the printed layers on the final strength of the object. Static tensile tests allowed the material constants of 316L steel in the form of dimensionally printed laminate to be determined. Tests were conducted on samples with different printed angles. The tests also covered the impact of the printing envelope on samples with the printing angles. Based on the determined material constants, simulation models for calculations using the finite element method were created. Furthermore, the study includes analytical and simulation calculations of plain laminate in order to verify the accuracy of the Composite Layup module in Abaqus CAE software. The study was summarized by compiling and commenting on the results obtained from the conducted research. Tests showed that there is a possibility of simulating the strength of the printouts from 316L steel using the FEM calculations. It was shown that the FEM model results are similar to those obtained in the tests. The calculated errors were from 3.6 to 14.4%. The linear model describes well the first part of the stress–strain curve, but in further research, it is strongly recommended that a proper and checked nonlinear anisotropic one is presented.

## 1. Introduction

Additive techniques known as 3D printing are developing dynamically these days because of their wide range of practical applications. Recent trends in, for example, organ or joint replacement have forced engineers to conduct research. Moreover, additive manufacturing allows the creation of complex geometric parts directly from computer-aided design (CAD) files, which cannot be produced by traditional manufacturing processes. The layer-by-layer build-up can reduce costs and material consumption. [1,2]

The history of additive manufacturing began 40 years ago. The first form of printing was rapid prototyping of samples/objects in plastic. It was used to help engineers show what they had in their minds. Nowadays, additive manufacturing is not only used for creating parts in plastic but also in other materials such as steel. Moreover, today’s technology allows different materials to be used while building a part. This process is called multiple material stereolithography.

The 3D printing process begins with a CAD project of printing samples. The next step is to export the CAD file to STL format. STL files were created by 3D systems Inc., which has become the standard for every 3D printer. Conversion from a CAD file to an STL file means that continuous geometry is interpolated to a header and small triangles. This type of file cannot transmit color, structure, or unit of measure. The user can impose parameters of conversion such as resolution, and linear or angular tolerances. Linear tolerance determines the quality of the net. The smaller the triangles, the more detailed the printed object, but also the bigger the size of the file. The influence of the linear tolerance is shown in Figure 1 [3].

The next step of the process is slicing the part. An exported STL file has to be opened in a Computer Aided Manufacturing (CAM) program, which replaces the continuous net of the triangles with discrete layers/steps and generates the paths of every layer for the head of the 3D printer. The thickness of the layers is chosen depending on the printed material. Screenshots from the KISSlicer v1.6 program of the layers generated by the slicer are shown in Figure 2. Red parts present the first printing layer known as the “skirt”. The grey parts present support, blue—envelope, and green—infill extrusion. When all the parameters of the printing are set, there is time to generate a G-code and send it to the 3D printer [5,6].

Additive manufacturing can be classified in, for example, terms of the material used for printing, such as polymers, polymers with fibers, composites, and metals [7]. Due to the development perspective and the enormous possibilities of 3D printing, which have developed a lot over the last ten years, there has been a need to make three-dimensional prints from metals. This can be undertaken by powder bed or by direct deposition. To metal 3D printing methods belong, for example, Selective Laser Sintering, Direct Metal Printing, Binder Jetting, Electron Beam Melting, and Fused Deposition Modeling. Direct Energy Deposition with WAAM is a method where the user may create the structure or repair it [8]. Moreover, the last of the above-mentioned methods can be used in 316L steel printing [3,6,9]. FDM is a great alternative to conventional metal 3D printing because of its low energy consumption, fast making, and low cost of printouts [10]. In [11], the authors introduced the 3D printing methods and showed the rapid increase in interest in printing nonflat elements. Step-by-step research was made on printing parameters’ influence on physical and mechanical properties, for example, the density of the probe. Very deep research on the printed composites made from at least two different materials as described in [12], where mechanical experimental test results were shown and extended with specimens of cross-section SEM images.

The Fused Deposition Modeling process is one of the additive processes where parts are formed by deposition of melted material and formed into layers. In FDM printers, the extruder head moves in an X-Y direction and the table moves in a Z direction. When the layer is ready the table goes down according to the layer thickness. In this method, there is sometimes a need to use support materials apart from the build materials in order to uphold parts until it is strengthened. A schematic representation of the FDM printing method process is presented in Figure 3. Most FDM printers use thermoplastic materials such as Acrylonitrile Butadiene styrene (ABS), but as was said before, there is also the possibility to use this method in printing from 316L steel in form of a composite [13,14]. According to [15], there is a need for investigations into connections between materials, process parameters, and mechanical properties in the FDM method. A great deal of research about the influence of printing parameters such as layer thickness, infill density, and support style had been conducted and presented in [16], but there is still a need to investigate the impact of the printing angle.

There are many articles that have considered the results of research on FDM-printed samples from polymer materials, which is not the main topic of this article. We are instead looking at 316L stainless steel, which is a popular material for many purposes, for example, medical or veterinarian implants [17]. In the available literature, the specimens from the mentioned material were investigated, but mostly they were received using methods that included laser melting or laser sintering. Melting process modeling [18], constitutive model preparation [19], and the evaluation of material parameters [20] are popular topics for laser-printed specimens.

The experimental test results for FDM 316L 3d printed specimens were presented in the literature [21]. The influence of the printing layer direction on mechanical properties, such as ultimate strength, was examined in [22], but these were experimental results only. A comparison of 316L specimens printed using the FDM and SLS processes was performed and the results were presented in [23]. The authors declare that the FDM method is cheaper and well suited for simple structures. The printed elements with the envelope tests were carried out in [24], where the authors noticed delamination in this type of specimen and suggested that the FEM models should be obtained.

However, there is still a lack of proper modeling of these kinds of elements; thus, we decided to use the model for laminated systems for the linear part of the examination results. This proposed model is a preliminary study, and it will become more comprehensive in our further works, where we will take into consideration the nonlinear behavior of the material also.

## 2. Materials and Methods

### 2.1. Experimental Methods

#### 2.1.1. Materials and Specimens

The material under investigation was an Ultrafuse 316L filament with a diameter of 1.75 mm ordered from BASF. The filament is a metal-polymer composite designed for 3D printing methods. It is suitable for printing metal elements on almost any 3D printer based on the FDM/FFF method. Filament Ultrafuse 316L consists of 80% powdered 316L steel bounded with a polymer that is removed during the postprocessing. 316L steel [Table 1] has found applications in, for example, surgical instruments, implants, and machine parts. Because of a need to quickly develop the prototype, BASF, in response to client needs, developed Ultrafuse 316L. Printing from this filament requires several steps. In the beginning, all of the printing parameters (layer thickness, scan speed, and extruder temperature) had to be set. When everything was prepared, then it was time to print green parts. That was the first step of printing from Ultrafuse 316L. Then, 3D-printed samples were outsourced to a debinding and sintering service for postprocessing. This processing was needed to remove the polymer from the part, with the finished metal part as the result. The debinding and sintering processes have to be performed using special devices. Sintered prints were then polished and prepared for tests. An example of 3D printing of an Ultrafuse 316L composite is shown in Figure 4 and Figure 5. The reader can see that printings from 316L steel have complicated geometry.

The subsequent shrinkage of the sample during sintering should be taken into account at the design stage, when the parts will be printed from Ultrafuse 316L composite. During sintering, the dimensions of the element shrink anisotropically. For this reason, the final dimensions of the project should be scaled by +120% in the direction of the *X*/*Y* axis and +126% in the direction of the *Z* axis.

#### 2.1.2. Printed Samples

Samples were printed from composite Ultrafuse 316L with different printing angles (lamination angles) 0°; ±30°; ±60°; 90°; and 90° with envelope. Dimensions of all tested samples were the same. Every sample had 27 layers, 0.122 mm thickness each. Dimensions of tested samples are summarized in Table 2. An example of tested samples is shown in Figure 6 in order to show their shape.

Tests were performed on samples with different lamination angles in order to estimate the influence of the stress–strain of the tested material. To average the results and ignore the effect of improper sample preparation, 55 samples overall were taken to be tested.

#### 2.1.3. Static Tensile Tests

Static tests were conducted using a testing machine INSTRON 3382 (see Figure 7) located in the laboratory of the Department of Materials Science and Engineering at Warsaw University of Technology. The feed rate was set at $4\frac{\mathrm{mm}}{\min}$ until the sample elongation was 4 mm. After reaching this value the speed changed to $20\frac{\mathrm{mm}}{\min}$. One strain gauge was used for the measurements, which was removed without stopping the test at an elongation of 4%.

#### 2.1.4. Probe’s Scans and Poisson’s Ratio

After the static tensile tests of all samples, there was a need to calculate Poisson’s ratio for printed 316L steel. Averaged dimensions of the specimens taken into the simulation is presented in the Table 3. Measurements were made on scans of the samples using Solidworks software (Figure 8 and Figure 9). The Poisson’s ratio was calculated from Formula (1).
(1)$$v=\frac{\varepsilon_{n}}{\varepsilon_{m}}=\frac{\frac{\Delta g}{g_{0}}}{\frac{\Delta l}{l_{0}}}$$
where $\varepsilon_{n,m}—$ normal and axial deformation.

### 2.2. Numerical Implementation

In order to perform analysis with the Finite Element Method, a simulation model was developed in the Abaqus CAE 6.6 program. Due to the differences in the dimensions of the cross-section of tested samples, the thickness and width of the samples were averaged in the FEM linear model and the values are given in the Table 3.

Based on the dimensions shown in Table 6, a conventional shell model of the sample (see Figure 10) was created and assigned Composite Layup—Figure 11.

Material created in Abaqus CAE responded to the printed 316L steel, and all of the needed parameters were taken from tensile static tests.

The idea of the simulation process was to reflect real test conditions. Therefore, boundary conditions (Figure 12) had to imitate machine jaws and the load had to be spread over the surface.

The next step of the simulation process was generating the mesh. In order to meet the conditions of a “good mesh”, the model had to be divided into parts (shown in Figure 13). The global size of a mesh was set to 1.5 mm, and it was proved that this size ensures accurate results in the shortest possible time. The elements highlighted in yellow did not meet the conditions of good mesh. The model was deemed to be properly prepared for the calculations.

During tensile strength tests, the displacements were measured by an extensometer with a measuring point spacing equal to 50 mm. This length had to be reflected in Abaqus in order to compare results from simulations and experimental research. It was performed by creating a path on the model (Figure 14).

## 3. Results

### 3.1. Experimental Methods

The tensile strength between the static and moving pistons was recorded during the tests. Graphs showing dependence between stress and strain were prepared based on the acquired data, separately, for each type of sample. A sample stress–strain graph is shown in Figure 15.

The data from the trials were collected in the form of the table. Machine software was able to find yield strength, tensile strength, and Young’s modulus, what was presented in the Table 4. Results significantly different from the others were marked red and not included in the calculation of the average values. Ten of the tested samples are shown in Figure 16.

Displacements in samples measured by the extensometer were read from raw data tables, which, due to their length, will not be mentioned in the following text.

When it comes to Poisson’s ratio, it was calculated from Formula (1).

Dimensions of samples with lamination angle 0° and 90° without envelope are summarized in Table 5 and Table 6. 

The next step of the experience was collecting all the averaged results in a table and then moving on to the Finite Element Method calculations and simulations. In Table 7 are shown all of the constants of printed 316L steel depending on the lamination angle.

### 3.2. Numerical Implementation

The FEM model was created based on printed 316L steel constants (Table 8) for the lamination angles 0° and 90°. Displacements in samples were calculated for angles ±30° (Figure 17) and ±60° (Figure 18) in order to compare the validity of the prototype parts to be printed.

### 3.3. Comparison of Displacement Results from the Experimental and Numerical Methods

Displacements in the direction of stretching were collected in Table 9 and Table 10.

## 4. Discussion

The conducted research and simulations allowed for the realization of the work’s goal, which was the construction and validation of simulation models of samples made from 316L steel by applying an additive technique.

Static tensile tests were carried out on the samples with lamination angles 0°, ±30°, ±60°, 90°, and 90° with envelope. The conducted research allowed for the determination of averaged yield strength, tensile strength, and Young’s modulus (Table 7). Poisson’s ratio for samples with lamination angles 0° and 90° were calculated based on the probe’s scans. The conducted research showed that the stiffness of printed steel 316L does not depend to a large extent on the printed angle. The values of Young’s modulus for each type of sample amounted to about 80–90 MPa. The yield points were about 120 MPa for samples printed without an envelope and almost 148 MPa for samples with an envelope. This means that additional layers at the edges of the printout delay the development of permanent plastic deformation. In summary, adding an outline to the printout strengthens its properties.

Referring to the tensile strength of the samples, the obtained results clearly show that the highest strength can be obtained by loading the printout along the fibers. Samples with a printing angle 0° achieved an average tensile strength of 470 MPa, which is more than three times that of samples with a printing angle 90°. This property should already have been taken into account for the part at the design stage to ensure the required resistance to later load action on the printout. The print envelope increased the tensile strength of the sample by more than double, thus confirming again the advantages of its use. Similar tensile strength tests were carried out in [26], where for the 316L specimen, the measured value was equal to 485 MPa.

The calculated Poisson’s ratio for the sample with the 0° printing angle was v=0.65. Research has shown that printed 316L steel stretched along the fibers has the desirable property of high plastic deformation (high tear resistance). As for the samples with a printing angle 90°, a Poisson’s ratio equal to zero means that the material broke faster than it deformed—between the printing layers.

The next step of the research was preparing a simulation model based on static tensile tests of samples with lamination angles 0° and 90° (Table 8). The displacement results from both methods were compared for samples with printing angles ±30° and ±60° (Table 9 and Table 10). The conducted research showed that in the range of linear deformations, the FEM models of printed elements gave results with a slight error. The relative error equal to 14% may have been caused by slight differences in the position of the extensometer during the tests or from an inaccurately determined Poisson’s ratio. In our research, we confirmed what was written according to [27], where the sample with 0° angles should have the highest strength. However, thanks to our FE models, engineers and researchers may easily be able to find the strength and deformation of specimens in situations where they were using other printing directions, which will improve the whole process of designing elements.

The aim of the work, which was to create simulation models of samples printed from steel 316L, was achieved, and constructed models reflect the actual behavior of the samples with a slight error. In order to reduce differences in the results, tests should be carried out on a larger number of samples with a wider range of sizes. The quality of the samples’ printing and the accuracy of their sintering turned out to be quite important. In addition to the experimental tests, all of the samples should be sintered during one process, which was not possible in our tests due to the size of the sintering chamber.

Research on 316L steel printouts should be continued and the FEM models should be extended to include a nonlinear range. Additionally, it is also worth carrying out experimental studies for nonlinear orthotropy.

The tests were carried out using the specified operating parameters of the machine. Subsequent tests should also take into account other (especially smaller) feed rates.

The provider of the Ultrafuse 316L composite emphasizes that non-slender prints will work better for this material, and because of this, elements of greater thickness should be tested.

The dispersion of the obtained Young’s modulus values for various printing angles of samples without an envelope is insignificant. When it comes to the strength of the printout, the tensile strength is the most important. Research on the construction of subsequent FEM models of 316L steel prints should focus on taking into consideration the breaking moment in response to the applied load.

To sum up the work, it should be stated that the goals have been achieved. The conducted analyses showed that further research is required in order to improve the prototyping capabilities of the models for printing, which will later be subjected to various loads and used in engineering structures such as the frame of a car, where the cost of the machining and preparing for the manufacturing process will be lowered [28].

## Figures and Tables

**Figure 1 materials-15-06244-f001:**
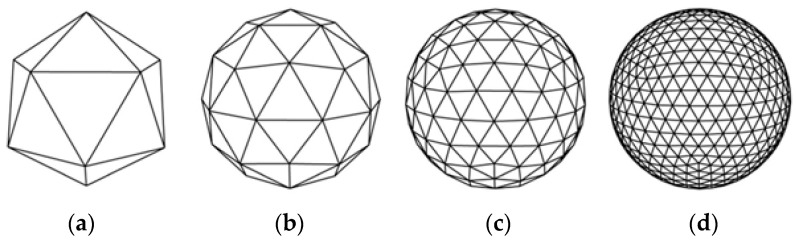
Converting a sphere to STL format depending on the given linear tolerance: from (**a**) the greatest linear tolerance, thru (**b**) and (**c**), to (**d**) the smallest linear tolerance [4].

**Figure 2 materials-15-06244-f002:**
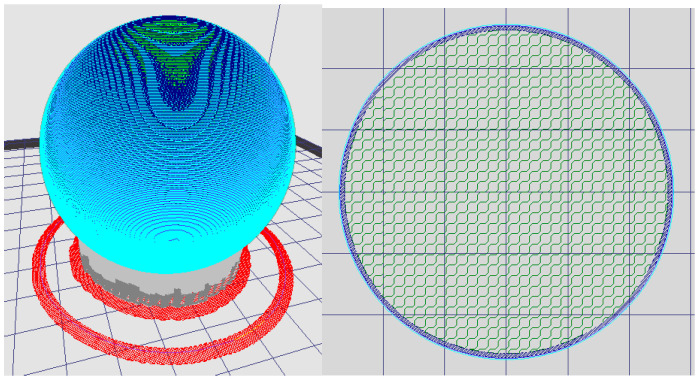
Paths generated in CAM program.

**Figure 3 materials-15-06244-f003:**
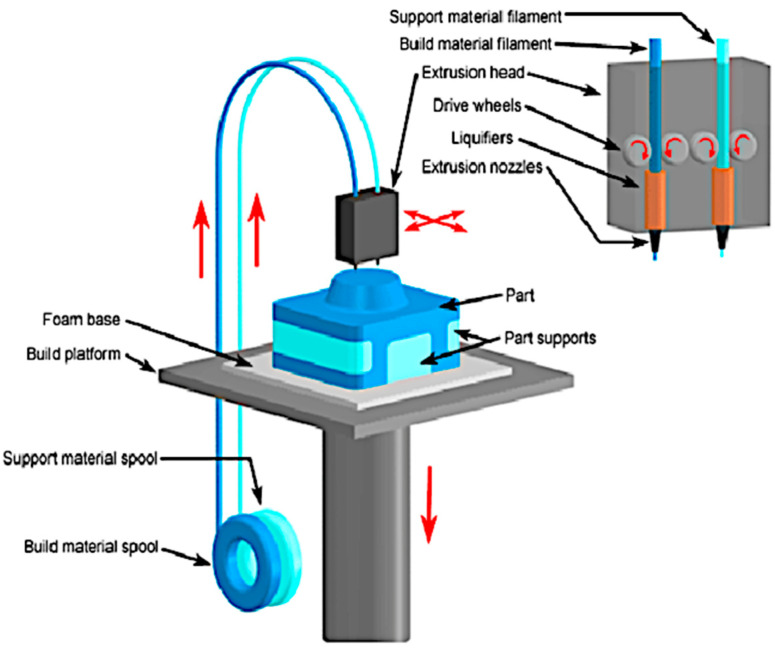
Schematic representation of the FDM printing method process [8].

**Figure 4 materials-15-06244-f004:**
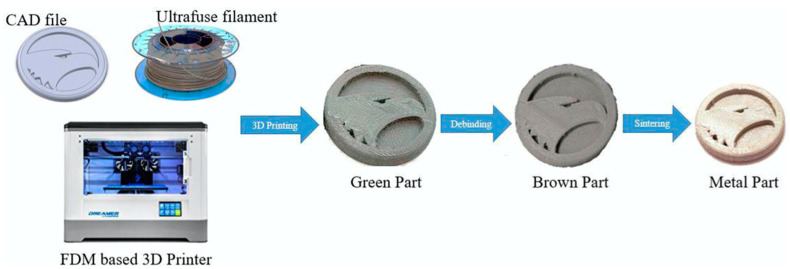
Schematic of Ultrafuse 316L printing process [23].

**Figure 5 materials-15-06244-f005:**
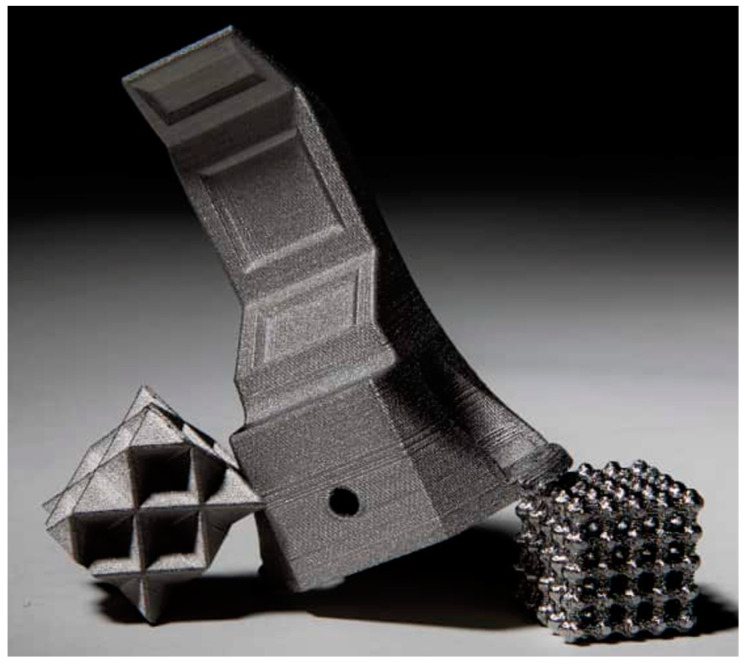
Elements printed from Ultrafuse 316 [25].

**Figure 6 materials-15-06244-f006:**
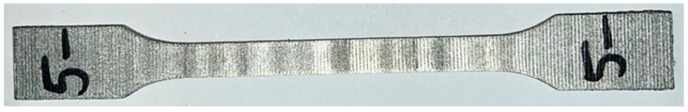
Example of tested samples (lamination angle 90°).

**Figure 7 materials-15-06244-f007:**
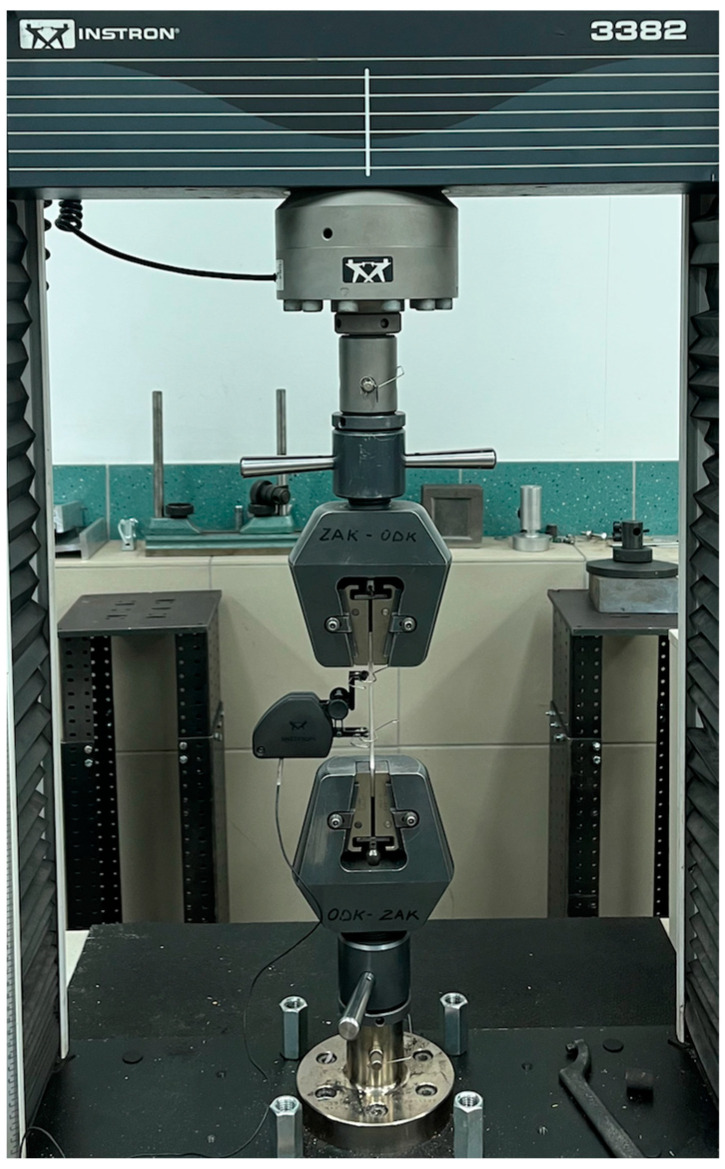
Sample during tests carried out on the Instron 3382 machine in Warsaw at WUT. The machine was made in the USA.

**Figure 8 materials-15-06244-f008:**
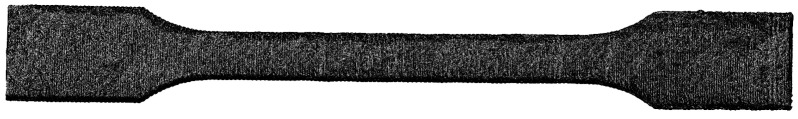
Scanned sample before static tensile test.

**Figure 9 materials-15-06244-f009:**
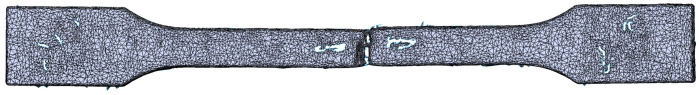
Scanned sample after static tensile test.

**Figure 10 materials-15-06244-f010:**
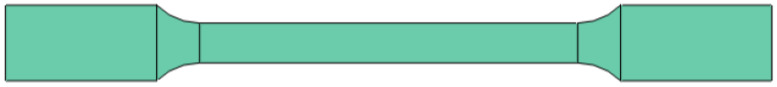
Model of sample created in Abaqus CAE.

**Figure 11 materials-15-06244-f011:**
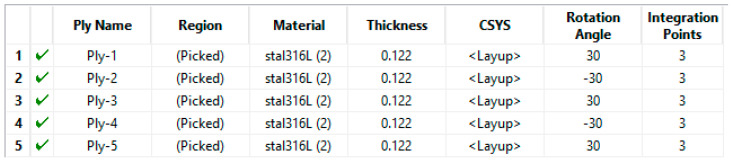
Part of Composite Layup assigned to the model.

**Figure 12 materials-15-06244-f012:**
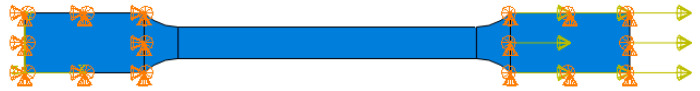
Boundary conditions and load.

**Figure 13 materials-15-06244-f013:**
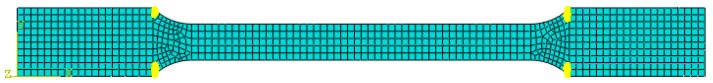
Mesh generated on the model of tested samples.

**Figure 14 materials-15-06244-f014:**
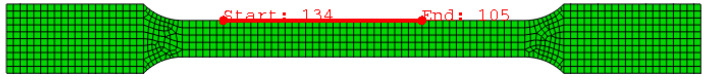
Path reflecting the extensometer.

**Figure 15 materials-15-06244-f015:**
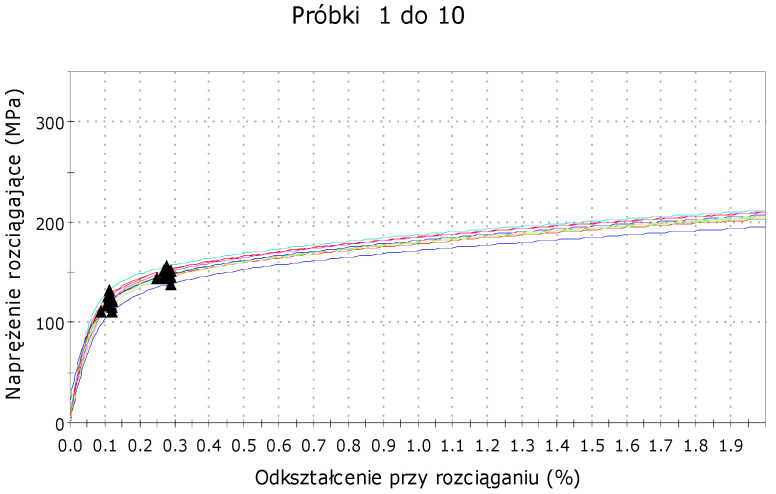
Stress–strain graph for samples with lamination angle 90° and with envelope.

**Figure 16 materials-15-06244-f016:**
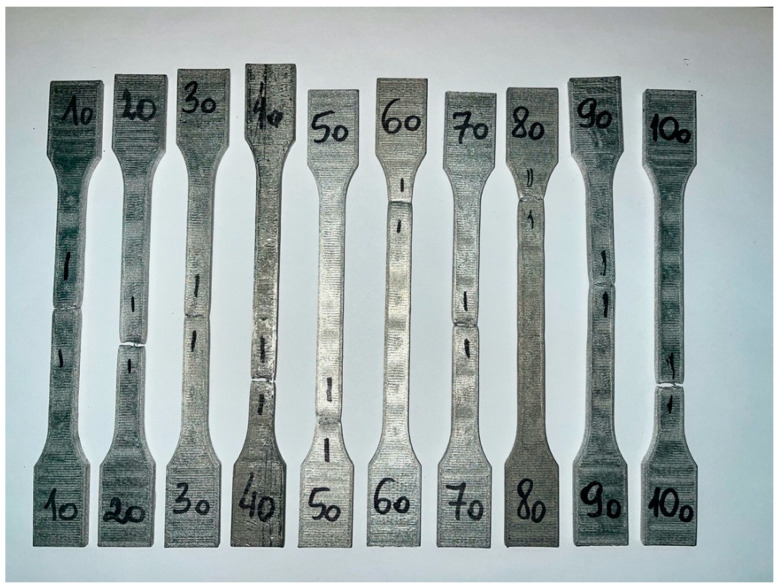
Tested samples with lamination angle 90° and envelope.

**Figure 17 materials-15-06244-f017:**
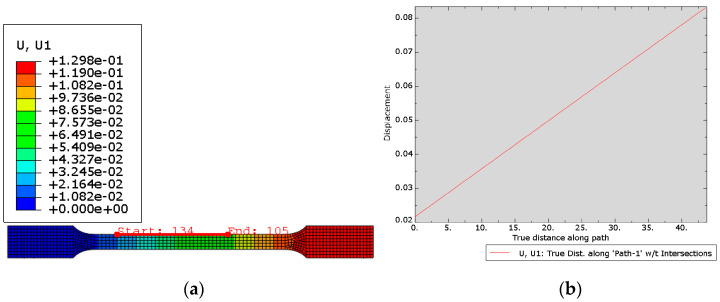
(**a**) Displacement map; (**b**) Plot from the path for sample with lamination angle ±30°.

**Figure 18 materials-15-06244-f018:**
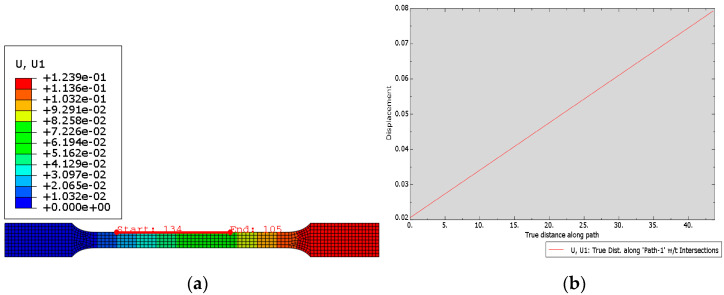
(**a**) Displacement map; (**b**) Plot from the path for sample with lamination angle ±60°.

**Table 1 materials-15-06244-t001:** Material properties of 316L steel [9].

E (GPa)	v (−)	$R_{m}\;\left(\mathrm{MPa}\right)$	$\varepsilon_{max}\;\left(-\right)$	$\rho \;\left(\frac{\mathrm{kg}}{m^{3}}\right)$
190–205	0.265–0.275	460–620	0.3–0.51	7870–8070

**Table 2 materials-15-06244-t002:** Dimensions of tested samples.

$\mathrm{Width}\;b_{0}\;\left(\mathrm{mm}\right)$	$\mathrm{Thickness}\;a_{0}\;\left(\mathrm{mm}\right)$	Number of Layers	Thickness of Layer (mm)	$\mathrm{Measuring}\;\mathrm{Length}\;l_{0}\;\left(\mathrm{mm}\right)$
7.9±0.3	3.3±0.1	27	0.122	75

**Table 3 materials-15-06244-t003:** Averaged dimensions of simulation model of tested samples with Composite Layup.

$\mathrm{Width}\;b_{0}\;\left(\mathrm{mm}\right)$	$\mathrm{Thickness}\;a_{0}\;\left(\mathrm{mm}\right)$	Number of Layers	Layer Thickness (mm)	$\mathrm{Measuring}\;\mathrm{Length}\;l_{0}\;\left(\mathrm{mm}\right)$
7.9	3.3	27	0.122	75

**Table 4 materials-15-06244-t004:** Data collected from static tensile strength for samples with lamination angle 90° and envelope.

	Width (mm)	Thickness (mm)	Yield Strength (0.2%) (Mpa)	Tensile Strength (Mpa)	Young’s Modulus (Mpa)
1	8.4	3.30	145.7	287.5	95 384
2	8.4	3.30	151.3	309.1	104 172
3	8.4	3.30	148.0	343.3	94 870
4	8.4	3.30	155.7	364.5	118 331
5	8.4	3.30	144.4	235.2	87 035
6	8.4	3.30	151.1	319.4	89 617
7	8.4	3.30	144.7	230.0	94 131
8	8.4	3.30	138.8	249.9	74 835
9	8.4	3.30	152.2	322.4	95 953
10	8.4	3.30	145.4	3.0	81 643
Average	8.4	3.30	147.7	295.7	93 597

**Table 5 materials-15-06244-t005:** Measurements results of samples with lamination angle 0°.

Sample Number	$\mathrm{Starting}\;\mathrm{Thickness}\;g_{0}\;\left(\mathrm{mm}\right)$	Final Thickness g (mm)	$\mathrm{Starting}\;\mathrm{Length}\;l_{0}\;\left(\mathrm{mm}\right)$	Final Length l (mm)	$\mathrm{Poisson}’s\;\mathrm{Ratio}\;\nu_{12}\;\left(-\right)$
7	3.2	2.5	75	114	0.64
8	3.2	2.5	75	112.9	0.66
9	3.2	2.5	75	112.4	0.66
				**Average**	0.65

**Table 6 materials-15-06244-t006:** Measurements results of samples with lamination angle 90° without envelope.

Sample Number	$\mathrm{Starting}\;\mathrm{Thickness}\;g_{0}\;\left(\mathrm{mm}\right)$	Final Thickness g (mm)	$\mathrm{Starting}\;\mathrm{Length}\;l_{0}\;\left(\mathrm{mm}\right)$	Final Length l (mm)	$\mathrm{Poisson}’s\;\mathrm{Ratio}\;\nu_{12}\;\left(-\right)$
1	3.2	3.2	75	86.3	0
7	3.2	3.2	75	86.5	0
8	3.2	3.2	75	83.2	0
				**Average**	0

**Table 7 materials-15-06244-t007:** Results collected from tensile strength tests.

Lamination Angle	$R_{pl}\;\left(\mathrm{MPa}\right)$	$R_{m}\;\left(\mathrm{MPa}\right)$	E [(MPa)]	$\mathrm{Poisson}’s\;\mathrm{Ratio}\;v_{12}\;\left(-\right)$
0°	119.1	469.15	80,216	$\nu_{12}=0.65$
±30°	128.6	449.6	92,656	−
±60°	121.7	306.9	86,572	−
90°	121.5	127.6	88,032	$\nu_{12}=0$
90° (with envelope)	147.7	295.7	93,597	−

**Table 8 materials-15-06244-t008:** 316L steel parameters necessary to carry out the FEM calculations.

$E_{1}\;(\mathrm{MPa})$	$E_{2}\;\left(\mathrm{MPa}\right)$	$G_{12}\;\left(\mathrm{MPa}\right)$	$v_{12}\;\left(-\right)$
80,216	88,032	24,308	0.65

**Table 9 materials-15-06244-t009:** Results for the sample with lamination angles ±30°.

	$\mathrm{Displacement}\;u_{x}\;(\mathrm{mm})$
Experimental method	0.0522
FEM analysis	0.061
Relative measurement δ (%)	**14.4%**

**Table 10 materials-15-06244-t010:** Results for the sample with lamination angles ±60°.

	$\mathrm{Displacement}\;u_{x}\;(\mathrm{mm})$
Experimental method	0.0559
FEM analysis	0.058
Relative measurement δ (%)	**3.6%**

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
