# Peer review of "Construction and Validation of Simulation Models of Samples Made from 316L Steel by Applying Additive Technique"

_materials, 2022, doi:10.3390/ma15186244_

Round 1
Reviewer 1 Report
Please kindly consider the attached file

Author Response
Thank you once again for the review
Please see the attachment

Reviewer 2 Report
The paper is well written and interesting for researchers working in the field of additive technologies.
At the end of the Abstract, add specific results that were obtained during the study described in the paper.
The Review part contains mainly general questions about 3D printing. However, it is necessary to expand the review of works directly on the paper's topic: to give a complete description of the state of the issue on the narrow problem that the paper is devoted to.
In general, the study Methodology is described in sufficient detail. Authors can indicate which microscope was used for metallographic studies and write in brackets the country and equipment manufacturers.
A large number of experimental results are in section 2 (Methods and Methodology), so it would be more logical to transfer them to section 3 (Results).
The paper would also be improved by bringing the microstructure of the printed products or the porosity values for different printing directions. This would allow a better understanding of the reasons for changing properties in different print directions.
In the Conclusions, more specific values of the obtained results should be given: indicate the deviation values between the calculated and actual results.
Also give values or percentage changes of properties in different print directions, if possible.
Author Response
Thank you once again for the review
Please see our response in the attachment

Reviewer 3 Report
In the manuscript “Construction and validation of simulation models of samples made from 316L steel by applying FDM additive technique”, authors used the simulation model for laminated systems through FDM technique. It contains some interesting findings. However, manuscript requires extensive major revision in following areas:
- Abstract is not written well. Authors must add some qualitative and quantitative findings in abstract.
- Keyword: Replace ‘3D printing in steel’ with “3D Printing”
- Explain different steps of figure 1.
- Line 68: In figure 3 was presented schematic form of FDM printing method. Rephrase this sentence.
- I am not satisfied with introduction section. Flow of the content is confussing. Modify it.
- It is necessary to add the literature of past studies. I didn’t observed any literature work in the introduction section
- Add more description techniques of AM. Following articles will be useful for this purpose: https://doi.org/10.1016/j.jmst.2018.09.002; https://doi.org/10.3390/app12105060; https://doi.org/10.1016/j.compositesb.2018.02.012
- Add the literature of recent work carried out by the researchers in FDM and elaborate it in more detail and clearly mention the research gap through that study. Below articles will be helpful for the same: https://doi.org/10.1016/j.polymertesting.2018.05.020; https://doi.org/10.1016/j.matpr.2019.09.078; https://doi.org/10.3390/polym13101587
- Improve the qualitu of figure 3
- Materials and Methods section is even more confusing. It is very confussing. Authors must avoid unnecessary texts an figures. Remove figure 5
- Line 123: Samples as it was said before……….. Rephrase the sentence
- Add the description of figure 6 in text. Why it is used ?
- Remove figure 7.
- In results and discussion section, compare the findings with published literature.
- It is necessary to give technical/scientific reasons for every findings.
- Referencing is very poor in text as well as in the list. Just see the reference number 1. It is not an appropriate way to give references. Authors must revise the entire list as per the journal requirement.
- Use latest references
Author Response

(The authors gave the same response as above.)

Round 2
Reviewer 1 Report
Good luck
Author Response
Dear Reviewer,
Thank You for the comment in the 2nd review.
We are very happy that you find well our work.
Kind regards
Radoslaw Nowak
Reviewer 2 Report
In general, my comments have been corrected. Thank you. But the "Conclusion" section disappeared somewhere.
Author Response
Dear Reviewer,
Thank you very much for your comments and for accepting our rearrange of the article in the present form.
The Conclusion section trully disappered because during the rearrangement of the paper we decided to merge it with the 4th section.
If you think that we should sepparate those two parts for clearness of the work, than we may do it.
Thank you once again for the review.
Kind regards
Radoslaw Nowak
Reviewer 3 Report
Accept in present form
Author Response
Dear Reviewer,
Thank You for the comment in the 2nd review form.
We are very happy that our article is accepted in the present form by you.
Kind regards
Radoslaw Nowak